# Temporal Assessment of Biofumigation Using Mustard and Oilseed Rape Tissues on *Verticillium dahliae*, Soil Microbiome and Yield of Eggplant

Lingbo Meng [1], Yuhang Zhang [2], Shaopeng Yu [1], Abiola O. Ogundeji [2], Shu Zhang [2] and Shumin Li [2,*]

1   School of Geography and Tourism, Harbin University, Harbin 150086, China
2   Resource and Environmental College, Northeast Agricultural University, Harbin 150030, China
*   Correspondence: shuminli@neau.edu.cn; Tel.: +86-45155191170

**Abstract:** *Verticillium dahliae* is a soil-borne pathogen that is difficult to eliminate, causes a severe reduction in the yield and quality of eggplant, and could be controlled through environmentally friendly biofumigation. However, the mechanisms of its effects on the dynamic changes of soil microbes is not clear. In this study, we examined the efficacy of two *Brassica* vegetables, mustard (*Brassica napiformis,* BFN) and oilseed rape (*Brassica napus,* BFC), as biofumigants to control *Verticillium dahliae* and observed their effects on the soil microbial community compared with chemical fumigants (CF) and a control (CN) in a greenhouse where eggplant was continuously cultivated for 12 years. Illumina MiSeq sequencing was used to analyse the soil microbiome. Real-time PCR was used to assay in order to estimate the soil abundance of *Verticillium dahlia*, and the glucosinolate concentration in biofumigants was determined using HPLC. Results showed that the BFN treatment had a significant biocidal effect on *V. dahliae* in the soil, decreasing its abundance by 64.74% compared to the control (CN). No significant difference was observed between BFC and CN treatments. Yield in the BFN treatment improved due to a lower disease incidence rate and disease index. Biofumigation had significant effects on the community structure and abundance of bacteria and fungi. The bacterial diversity of the BFN treatment was significantly lower than those of the other treatments, and beneficial bacterial community relative abundance, such as *Flavobacteria*, *Pseudomonas* and *Bacillus*, increased after the soil was biofumigated for 15 days. However, the temporal difference in bacterial and fungi structure among the different treatments was reduced with the development of the eggplant growth stage. The *Mortierella*, *Altemaria* and *Chaetomium* genera were significantly abundant across all treatments. Soil enzyme activities were also improved. In summary, biofumigation using mustard was efficient in controlling eggplant Verticillium wilt due to changes in the soil bacterial composition at the early eggplant growth stage; the initial conditions in the soil bacterial community are a key determinant of what is going to happen after Brassica biofumigation, which could be considered a practical addition to integrated pest management for the reduction of soil-borne pathogens.

**Keywords:** biofumigaiton; eggplant; glucosinolates; microbial community; *Verticillium dahliae*

## 1. Introduction

Eggplant (*Solanum melongena* L.) is the fifth most economically vital solanaceous crop after potato, tomato, pepper, and tobacco [1]. As stated by the Food and Agriculture Organization of the United Nations, China is the world's largest eggplant producer (28 Mt per annum) where domestic production accounts for 70% of global eggplant production [2]. *Verticillium* wilt, which is caused by the soil-borne fungal pathogen, *Verticillium dahliae* Kleb., is one of the most destructive diseases of eggplant [3,4] and occurs in a wide range of economically significant solanaceous crops such as tomato, potato, etc. The disease affects more than 400 plant species in several families and causes vascular wilt and yield losses, especially in solanaceous crops [5]. Generally, farmers utilize synthetic

fumigants to control soil-borne pathogenic microorganisms. However, these also kill non-target microorganisms in the soil, such as harmless bacteria [6]. Due to the adverse effects of chemical disinfection and the noticeable prevalence of *V. dahliae,* it is necessary to find an ecofriendly and effective way to control *V. dahliae*.

Biofumigation refers to incorporating glucosinolate-containing plant parts into the soil to control soil-borne plant pathogens [7]. This approach utilizes the release of volatile compounds, such as isothiocyanates (ITCs), by the enzymatic hydrolysis of glucosinolate to kill pathogenic bacteria [8]. A study showed that Brassicaceae plants have been used in rotations and green manures to control verticillium wilt [9,10]. These methods have shown that continuous cropping soils amended by biofumigation have a suppressive effect on a range of soil-borne diseases, such as potato diseases caused by *Rhizoctonia solani* [11]. Moreover, some studies report effective control through the application of Indian mustard extract, which severely reduced the growth of the majority of *Monilinia fructicola* and *Botrytis cinerea* mycelium growth and conidia germination, respectively [12,13]. Hence, biofumigation could be considered a 'natural' alternative to chemical fumigation to control a variety of soil-borne diseases [14]. ITCs released from glucosinolate compounds may have an impact not only on soil-borne plant pathogens but also on the beneficial soil microflora [15]. Cohen et al. [16] reported alterations in communities of both pathogenic and saprophytic soil microbes after a *Brassica napus* seed meal application to orchard soils. Some other studies showed that the proportions of some beneficial soil microorganisms were improved after the soil was amended by broccoli residue [17,18]. The reason behind this effect might be the higher tolerance of the biocontrol microbe against the Brassica soil amendment decomposition products compared to phytopathogens [19,20]. However, few studies about the temporal assessment of soil microbes affected by biofumigation have been reported since the nutrient was added after the brassica plants were incorporated into the soil. Soil microbial composition is generally considered to be a soil quality indicator [21], and transformations of major organic constituents occur by soil microbes [22]. Soil enzyme activities, which are mainly believed to be of microbial origin [23], would be considered critical for the maintenance of microbial metabolisms. Except for that the content and composition of glucosinolates in different Brassica crops vary greatly. It is necessary to verify whether the effects of biofumigation oncontrol soil-borne plant pathogens affected by Brassica crops that contain different levels of glucosinolate.

However, there is a dearth of information on the biofumigation of crops using plant tissues of different glucosinolate levels, their suppressive effects on *Verticillium dahlia*, and the consequent temporal assessment variation in soil microbial composition and soil enzyme activities, especially in continuous eggplant cropping systems. Therefore, the aims of the study were to (i) evaluate the effects of biofumigation on verticillium wilt disease in eggplant and the yield of eggplant (ii) compare the effects of different glucosinolate concentrations containing crops in the biofumigants used, and (iii) assess the temporal effects of the biofumigation materials on the microbial community structures.

## 2. Materials and Methods

### 2.1. Experimental Design

This experiment was conducted in a greenhouse at Harbin Horticultural Institute, Heilongjiang Province, China (45°45′ N, 126°38′ E) where eggplant has been continuously cultivated in a single growing season due to climatic conditions for over 12 years. The disease incidence rate of *Verticillium* wilt on eggplants in this greenhouse was estimated to be about 60 percent. Preplanting soil properties at depths from 0–20 cm were determined: total C = 20.31 g kg$^{-1}$; total N = 1.41 g kg$^{-1}$; available $P_2O_5$ = 829.5 mg kg$^{-1}$; available $K_2O$ = 458.2 mg kg$^{-1}$; pH = 6.95.

Two kinds of brassica crops (mustard and oilseed rape) containing varying glucosinolate concentrations were selected as biofumigation materials to use in rotation with eggplant. The experiment layout was four treatments with three replications in a randomized complete block design. The treatments were mustard as fumigation materials (BFN),

oilseed rape as fumigation materials (BFC), chemical fumigation (CF), and the continuously cropped eggplant as control (CN). Each bed was 5 m (width) $\times$ 2 m (length) with a 50 cm interval between two vegetable beds and 1 m interplots to avoid contact between different treatments. Mustard (*Brassica napiformis* var. ErDao mei.) and oilseed rape (*Brassica napus* L. WoGuan-2) were planted on 18 August 2015 and harvested on 16 October 2015. The harvested crop tissue was chopped into pieces and mixed thoroughly into 20 cm depth soil by a rotavator. On the same day, some fresh samples were also taken to measure glucosinolate levels. Meanwhile, beds of mustard and oilseed rape plants (550 g) were taken at random and were divided into leaf and root for measuring glucosinolate content. Meanwhile, beds of CF treatment were fumigated with 50% carbendazim WP for 1.5 g m$^{-2}$ (Jiangsu Lanfeng Biological Chemical Co., Ltd., Xuzhou, China). Each plot was irrigated to about 75% soil capacity, and mulched using plastic to fumigate soil [24] for 20 days. Eggplant (*Solanum melongena* L. LongZa-9) were transplanted on 6 May 2016, soil samples (0–20 cm) were taken from ten points in each treatment according to the staggered grid method [25] using a soil auger 15 days after transplanting and eggplant fruiting stage. Soil samples were bulked according to treatments and immediately put into a −4 °C ice box and stored at −80 °C in an ultra-low temperature freezer.

### 2.2. Real-Time PCR Assay to Estimate Soil Abundance of Verticillium dahliae

Fresh soil samples weighing approximately 500 mg were placed in 2 ml DNA extraction tubes for the Fast DNA Spin Kit for Soil (MP Biomedicals, USA), frozen in liquid nitrogen for 2 min, and then put in a water bath (70 °C) for 2 min [26]. These steps were repeated for three cycles before the manufacturer's instructions were carried out. According to Bilodeau et al. [27], an alignment of 65 IGS sequences of different *Verticillium* spp. was used to design species-specific PCR primers and probes for *V. dahliae*. The software IDT SciTools Oligo Analyzer 3.1 (Integrated DNA Technologies Inc., Coralville, IA, USA) was used for primer design. Vd-F929-947 (CGTTTCCCGTTACTCTTCT) and Vd-R1076-1094 (GGATTTCGGCCCAGAAACT) were chosen to be *V. dahliae* assay primers. Vdhrc FAM (CACCGCAAGCAGACTCTTGAAAGCCA) was selected to be the *V. dahliae* probe. Quantification of the pathogen was determined using the *V. dahliae* quantification molecular assay. Furthermore, abundance was expressed as mean *V. dahliae* gene copies per dry weight (DW) gram of soil.

### 2.3. Illumina MiSeq Sequencing for Analysis of Soil Microorganism Abundance

DNA library prep was carried out using an ABI Gene 9700 (Applied Biosystems, Waltham, MA, USA). The PCR experiments used a Trans Start Fastpfu DNA Polymerase reaction system. The V4-V5 region of the 16S RNA bacterial gene was amplified using PCR with primers 338F (5′-ACTCCTACGGGAGGCAGCAG-3′) and 806R (3′ TACHVGGGTWTCTAAT-5′), where the barcode is an eight-base sequence unique to each sample. A 3 min denaturing at 95 °C was first carried out. For the second part, 27 cycles of denaturing for 30 secs at 95 °C were completed, annealed for 30 s at 55 °C and extended for 45 s at 72 °C. Third part involved a final extension for 10 min at 72 °C.

Illumina MiSeq sequencing was used to study microbial community variations in the soil samples. PCR products were purified using an Axy Prep DNA Gel Evulsion Kit (Axygen Company, Union City, CA, USA) Gel Extraction PCR product, eluted using Tris-HCl and quantified using Quanti Fluor TM-ST (US). Purified amplicons were sequenced as stated by Li et al. [19]. The raw reads were deposited into the NCBI Sequence Read Archive database.

Shannon–Weaver and Rarefaction data were created using mothur. Demultiplexed raw files were filtered by QIIME (version 1.17). Operational taxonomic units (OTU) clusters were done with a 97% similarity cutoff by USEARCH (version 7.1 http://drive5.com/uparse) (accessed on 10 November 2018), and single chimeric sequences were identified and removed by UCHIME (http://drive5.com/usearch/manual/singletons.html (accessed on 12 November 2019). An RDP classifier was used to sequence the taxonomy of the 16S rRNA against the 16S rRNA database using a probability limit of 70% [28].

### 2.4. Investigating Morbidity and Yield of Eggplant

Eggplant yield and morbidity were studied every two days from the beginning of the fruit setting period on 29 June 2016 to the final period of fruit on 4 August 2016. When the incidence rate reached 10%, the number of diseased plants and the disease severity were investigated and recorded in detail.

The Verticillium wilt disease grading standards according to Ogundeji et al. [3] were used

$$\text{Incidence rate } (\%) = \left( \frac{\text{number of diseased plants}}{\text{total number of plants in the survey}} \right) \times 100$$

$$\text{Disease Index} = \sum \left[ \frac{(\text{Number of infected plants at all levels } \times \text{ the corresponding grade level})}{(\text{The total number of plants investigated} \times \text{ the highest grade level})} \right] \times 100$$

### 2.5. Glucosinolate Concentration Analysis

Fresh mustard and oilseed rape plants were freeze dried ($-40\ ^\circ$C) and grounded. Glucosinolates contained in biofumigation materials were analysed according to [29]. Sinigrin was used as an internal standard. Sinigrin was bought from Sigma company. Its purity is 99%. Each sample was duplicated two times. Two-hundred microliters of a 5 mM stock solution of sinigrin in methanol was added to one of the duplicates just before the first extraction as internal standard. Mass spectrum was used to identify individual glucosinolate in standard reference materials of oilseed rape that enriched glucosinolate composition. Individual glucosinolate of biofumigation materials was measured by comparison of its retention time with that of individual glucosinolate in standard reference materials. HPLC Agilent 1200 (Agilent, Santa Clara, CA, USA), Novapak $C_{18}$ (250 mm $\times$ 4 mm, the grain diameter of 5μm) was used for analysis. The glucosinolate concentration was calculated using the response factor of each compound relative to sinigrin. GSs concentration was expressed in GSs micromoles per gram sample of dry powder ($\mu$molg$^{-1}$ DW) [30].

### 2.6. Soil Enzyme Activities

Soil urease activity was determined by colourimetry, expressed as $NH_4^+$ mg·g$^{-1}$·24 h$^{-1}$. Invertase activity was measured as reported by Ohshima et al. [31] using 15 ml of 80 g·kg$^{-1}$ sucrose as a substrate.

### 2.7. Statistical Analysis

SPSS software (IBM SPSS Statistics v19) was used for statistical analysis. The limit was set at $p \leq 0.05$ and value of p less than 0.05 was regarded as statistically significant. Values are presented as mean $\pm$ standard deviation. The statistical significance of data comparisons was done by analysis of variance (ANOVA, one-way) and means separated by Duncan's multiple range test. The NMDS analysis in this study was calculated using the Bray–Curtis dissimilarity matrix from the OTU table which assessed soil bacteria and fungi communities and structures.

## 3. Results

### 3.1. Variation of Soil Verticillium dahliae Abundance and Yield Improvement Affected by Biofumigation

Verticillium wilt was observed in all treatments (Table 1). There were about $35.04 \times 10^5$ g$^{-1}$ *V. dahliae* gene copies g$^{-1}$ in the control treatment, which was significantly higher than the other treatments (Table 1). *V dahliae* gene copies in the BFN treatment soil had the least abundance and were significantly lower than the other treatments.

**Table 1.** Effects of treatments on the abundance of *Verticillium dahlia* and yield improvement.

| Treatments | Number of *V. dahlia* ($\times 10^5$ Gene Copies g$^{-1}$ DW) | Incidence Rate (%) | Disease Index | Disease Prevention (%) | Yield ($\times 10^3$ kg·hm$^{-2}$) |
|---|---|---|---|---|---|
| CN | 35.04 ± 2.9 a [1] | 41.35 ± 1.3 a | 14.54 ± 0.4 a | - | 18.78 ± 0.2 b |
| BFN | 3.74 ± 0.2 cd | 25.19 ± 1.2 bc | 8.82 ± 0.3 c | 39.32 ± 2.1 a | 25.43 ± 1.0 a |
| BFC | 10.61 ± 0.8 b | 29.73 ± 1.6 b | 11.94 ± 0.2 b | 17.93 ± 1.6 b | 20.23 ± 1.7 b |
| CF | 5.22 ± 0.5 c | 24.12 ± 1.5 c | 8.45 ± 0.1 c | 41.89 ± 0.9 a | 23.94 ± 0.1 a |

CN = Control, BFC = Oilseed rape fumigation, CF = Chemical fumigation, BFN = Mustard fumigation. [1] Mean values (n = 3) followed by the same letters in the same column indicate no significant difference between different treatments ($p \leq 0.05$).

The lowest verticillium wilt incidence rate and disease index were observed in the BFN treatment, which was significantly lower than the BFC and CN treatments but not lower than CF treatment. Disease prevention between the BFN and CF treatments did not have a significant difference but was significantly higher than the BFC treatment. The eggplant yield in the BFN treatment was 35.41% and 25.70% higher than the control and BFC treatments, respectively, but was not significantly different compared with the CF treatment. Yield improvement was also observed from the BFC treatment.

*3.2. Soil Microbial Diversity Affected by Biofumigation*

Different soil fumigation had a significant influence on soil bacterial and fungi alpha diversities. At 15 days after eggplant transplanting, the soil bacteria Shannon index in the BFN treatment was significantly lower than the other treatment ($p \leq 0.05$). No significant difference was found among the BFC, CF and CN treatments (Figure 1a). For the fungi community, the Shannon index in the BFN treatment was also significantly lower compared with the CF and BFC treatments (Figure 1b). At the eggplant fruiting growth stage, both soil bacterial and fungi Shannon indices had no significant difference across all the treatments (Figure 1c,d).

To further clarify phylogenetic similarities, nonmetric multidimensional scaling (NMDS) based on Bray–Curtis and UniFrac distance was used to compare betadiversity patterns of microbial communities from the samples of treatments (Figure 2). The stress values of 0.108 and 0.022 showed how well the ordination summarized the observed distances among the samples for bacteria and fungi communities. The NMDS analysis indicated that the microbial communities in the different treatments were obviously separated in different quadrants 15 days after eggplant transplanting, which suggests that their bacterial structures are different (Figure 2a). The fungi community showed closer similarities with bacterial structures among the treatments (Figure 2b). However, NMDS analysis results at the eggplant fruiting stage were obviously different from those at 15 days. The NMDS analysis indicated that there are obvious intersections for fungi communities in the different treatments (Figure 2d). The bacteria communities in the CF and CN treatments tended to be similar, while the bacteria communities in the BFC and BFN treatments were still different at the fruiting stage (Figure 2c).

*3.3. Soil Beneficial Microbial Composition Affected by Biofumigation*

The relative abundance of bacterial and fungi results at the genus level showed a difference in composition at 15 days after transplanting and the eggplant fruiting stage. At 15 days after transplanting, beneficial soil bacteria of the eggplant samples in each treatment were *Flavobacteria, Arthrobacter, Bacillus* and *Pseudomonas* (Figure 3a). *Flavobacteria and Pseudomonas abundance in the BFN treatment was ten times that of the CN treatment.* However, there was a noticeable reduction in genera *Flavobacteria, Arthrobacter, Bacillus and Pseudomonas,* and no significant difference was found across the treatments at the fruiting stage while *uncultured_Anaerolineaceae* increased (Figure 3c). Similar trends were observed in the fungi community at 15 days after transplanting; soil fungi of the eggplant in each treatment were *Alternaria, Chaetomium* and *Cladosporium* (Figure 3b). The reduction of the

genus *Mortierella* was greater in the BFN treatment (Figure 3d). There was no significant difference for these fungi at the fruiting stage.

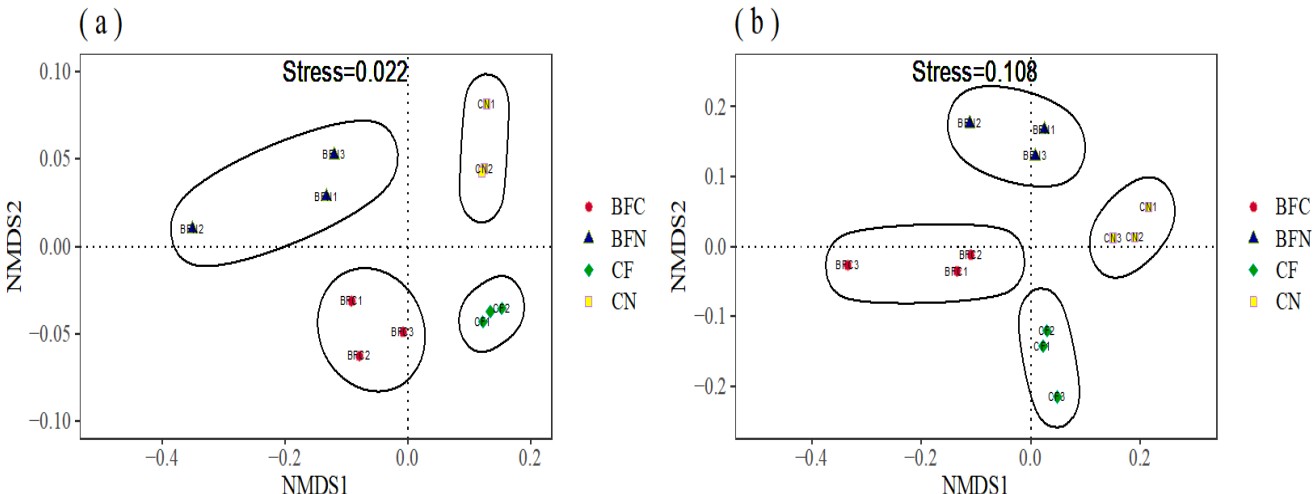

**Figure 1.** Shannon indices of soil bacteria and fungi 15 days after transplanting (**a**,**b**) and fruiting stages (**c**,**d**), respectively ($p \leq 0.05$).

**Figure 2.** *Cont.*

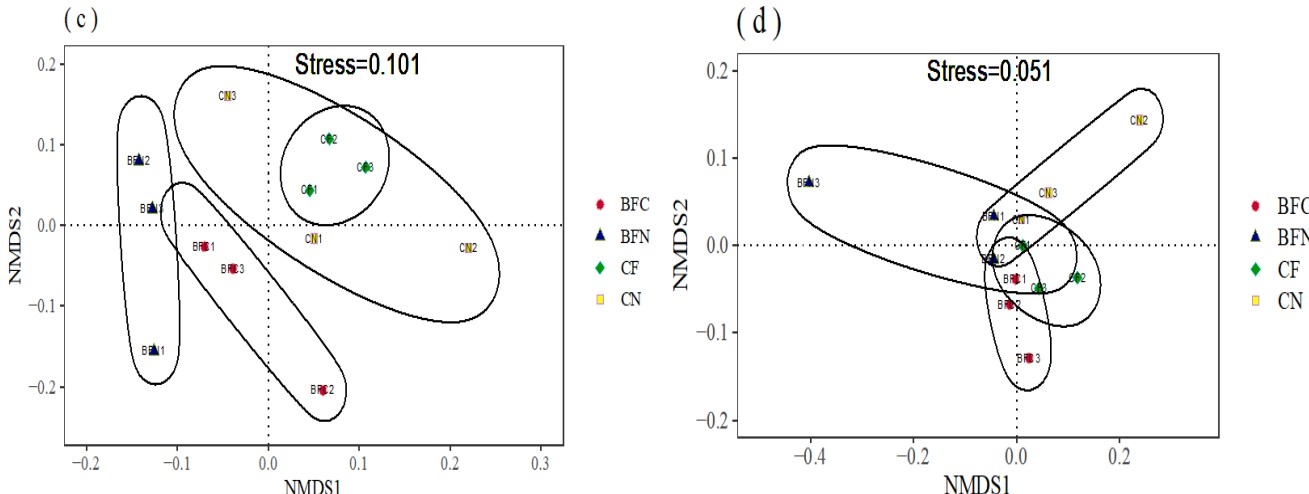

**Figure 2.** Nonmetric multidimensional scaling (NMDS) of bacteria and fungi community structure at 15 days after transplanting (**a**,**b**) and fruiting stages (**c**,**d**) based on Bray–Curtis dissimilarities. Points of different colours or shapes represent samples of different groups.

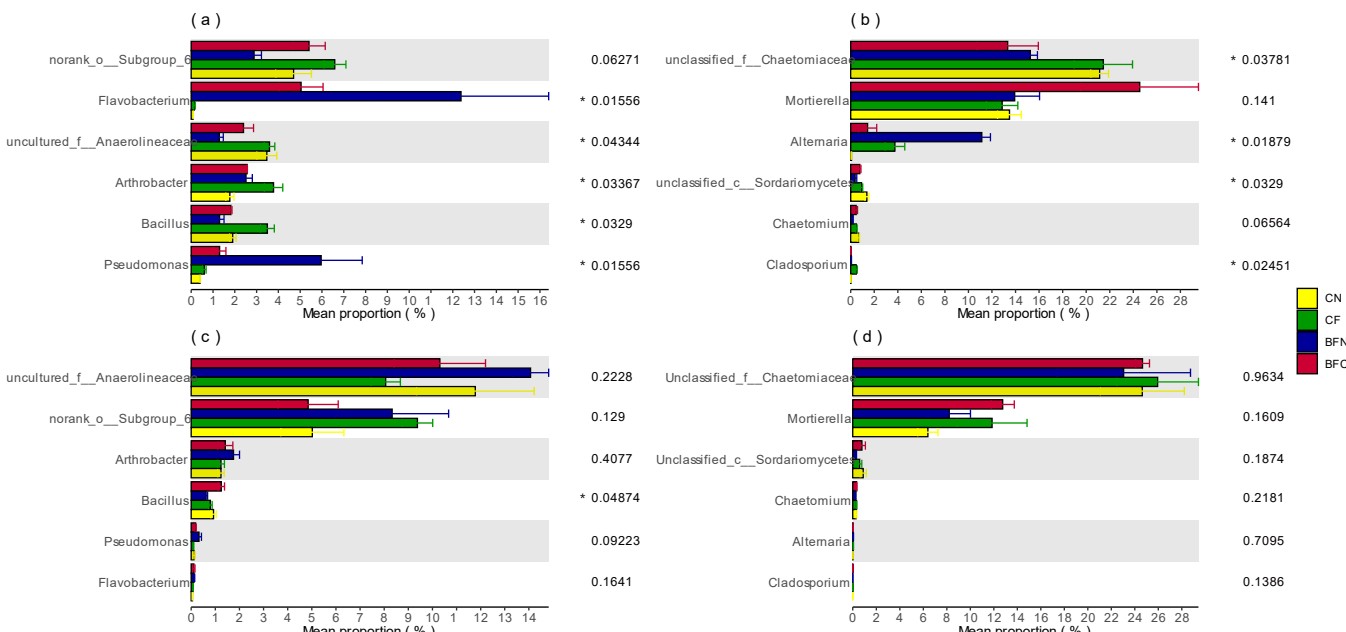

**Figure 3.** Relative abundance of some selected beneficial soil microbes 15 days after transplanting (**a**,**b**) and fruiting stages (**c**,**d**) for bacteria and fungi, respectively ($p \leq 0.05$). The bars in each column are standard errors. Means value (n = 3), * denotes significant difference at $p \leq 0.05$.

### 3.4. Glucosinolate Profiles in Tissues of the Biofumigation Materials

Sixteen GSs (glucosinolate compounds) were identified from both mustard and oilseed rape roots (Table 2). A significant difference was observed in glucosinolate concentration and the composition between mustard and oilseed rape (Table 2). The total GS content in the mustard roots was 34.75 µmolg$^{-1}$ dry weight (DW), of which, 95.31% corresponded to aliphatic GS, 3.65% to indolyl GSs and 1.04% to benzenic GSs. Compared with mustard, the GS content in oilseed rape was 6.55 µmolg$^{-1}$ DW, nearly one-fifth of that in mustard roots, and consisted of 91.91% aliphatic GSs, 3.51% indolyl GSs and 4.58% benzenic GSs.

**Table 2.** Glucosinolate concentration ($\mu mol \cdot g^{-1}$ DW) in the tissues of different biofumigation materials.

|  | Trivial Name | Chemical Name | Mustard | Oilseed Rape |
|---|---|---|---|---|
| *Aliphatic GS* | progoitrin | 2-hydroxy-3-butenyl GS | 0.76 ± 0.03 a [1] | 0.21 ± 0.02 b |
|  | pi-progoitrin | 2-hydroxy-3-butenyl GS | 0.32 ± 0.02 a | 0.29 ± 0.02 a |
|  | sinigrin | 2-propenyl GS | 29.56 ± 0.41 a | — |
|  | glucoraphanin | 4-methylsulfinylbutyl GS | 0.10 ± 0.03 b | 0.27 ± 0.01 a |
|  | gluconapoleiferin | 2-hydroxy-4-pentenyl GS | 0.52 ± 0.07 a | — |
|  | glucoalyssin | 5-methylsulfinylpentyl GS | 0.02 ± 0.01 b | 0.47 ± 0.03 a |
|  | gluconapin | 3-butenyl GS | 1.53 ± 0.02 b | 4.21 ± 0.23 a |
|  | glucoerucin | 4-methylthiobutyl GS | 0.16 ± 0.02 a | 0.02 ± 0.01 b |
|  | Glucobrassicanapin | 4-pentenyl GS | 0.11 ± 0.01 b | 0.38 ± 0.01 a |
|  | 5-methyl GS | 5- methyl GS | 0.04 ± 0.01 c | 0.17 ± 0.01 a |
| *Indolyl GS* | 4-hydroxyglucobrassicin | 4-hydroxyindol-3- methyl GS | 0.81 ± 0.04 a | 0.02 ± 0.00 b |
|  | glucobrassicin | Indol-3-methyl GS | 0.01 ± 0.003 a | 0.03 ± 0.006 b |
|  | 4-methoxyglucobrassicin | 4-methoxyindol-3- methyl GS | 0.05 ± 0.006 b | 0.16 ± 0.01 a |
|  | neoglucobrassicin | 1-methoxyindol-3- methyl GS | 0.40 ± 0.006 a | 0.02 ± 0.006 c |
| *BenzenicGS* | glucotropaeolin | benzyl GS | 0.05 ± 0.06 a | 0.09 ± 0.01 a |
|  | gluconasturtiin | 2-phenethyl GS | 0.31 ± 0.03 b | 0.21 ± 0.03 b |

[1] Mean values (n = 3) followed by the same letters in the same column indicate no significant difference between different treatments ($p \leq 0.05$). DW means dry weight.

Additionally, the concentration of individual GSs varied greatly in the plant tissues. In mustard, 2-propenyl (sinigrin) was found as the main glucosinolate in all groups. It was measured at 29.56 $\mu mol/g$ DW, 92% of the total GSs of the tissues. However, no sinigrin or gluconapoleiferin was found in the oilseed rape. The primary aliphatic GS in oilseed rape was gluconapin, followed by glucoalyssin and glucobrassicanapin. Despite the differences in GS content, the benzenic GS content in both *Brassica* crops was similar.

*3.5. Soil Enzymatic Activities Affected by Biofumigation*

The soil enzyme activities in biofumigated soil were significantly different from those in the CF- and CN-treated soil samples. Enzyme activity was found to be greater in the early fruit stage (Table 3). The soil urease activity in the BFN and BFC treatments were significantly different at the three growth stages, and the difference was more apparent in the early fruit stage. Compared with the BFN treatment, urease activity in the BFC and CF treatments decreased by 5.41% and 25.81%, respectively. The soil invertase activity in the BFN treatment was significantly higher than that in the BFC treatment during flowering and early fruiting. However, there was no significant difference in activity during the full fruit period. In the early fruit period, invertase activity in the BFN treatment increased by 36.15% compared to activity in the CN treatment and 32.10% compared to activity in the CF treatment.

**Table 3.** Effect of different treatments on soil enzymatic activities at different eggplant growth stages across the treatments.

| Treatment | Urease ($NH_3$-N mg·g$^{-1}$) | | | Invertase (Glucose mg·g$^{-1}$·24 h$^{-1}$) | | |
|---|---|---|---|---|---|---|
|  | Flower Stage | Early Fruit Stage | Full Fruit Stage | Flower Stage | Early Fruit Stage | Full Fruit Stage |
| CN | 0.33 ± 0.002 c [1] | 0.29 ± 0.001 d | 0.23 ± 0.002 d | 12.85 ± 0.177 d | 12.09 ± 0.143 c | 10.89 ± 0.116 b |
| BFN | 0.38 ± 0.004 a | 0.39 ± 0.002 a | 0.28 ± 0.002 a | 14.94 ± 0.130 a | 16.46 ± 0.106 a | 12.09 ± 0.124 a |
| BFC | 0.35 ± 0.005 b | 0.37 ± 0.004 b | 0.27 ± 0.002 b | 14.33 ± 0.186 b | 15.03 ± 0.082 b | 11.89 ± 0.130 a |
| CF | 0.34 ± 0.003 bc | 0.31 ± 0.004 c | 0.24 ± 0.002 c | 13.47 ± 0.124 c | 12.46 ± 0.117 c | 11.13 ± 0.129 b |

CN = Continuous cropping, BFC = Oilseed rape fumigation, CF = Chemical pesticide fumigation, BFN = Mustard fumigation, [1] Mean values (n = 3) followed by the same letters in the same column indicate no significant difference between different treatments ($p < 0.05$).

## 4. Discussion

### 4.1. Biofumigation Could Control Verticillium dahliae

In their tissue, Brassica vegetables contain different varieties and quantities of glucosinolates. These compounds co-occur with other enzymes; for instance, myrosinase acts as a catalyst to hydrolyse the β-D-thioglucopyranoside bond, breaking it down into a variety of other biologically active products among which are the main isothiocyanates. The Brassica vegetable residue treatments used in this study had a noticeable effect on the test soil pathogen. Both treatments reduced the *Verticillium dahliae* abundance by at least 60% (Table 1). This reduction in abundance is believed to be caused by a reduction in the formation of microsclerotia, consequently reducing the pathogenicity and virulence of the pathogen under the effect of the brassica vegetable residues. The present study observed that biofumigation treatments reduced the abundance of *Verticillium dahliae* in the soil, lowered disease incidence in the eggplant and increased yield (Table 1). This result agrees with other studies that reported that broccoli residue amendment effectively reduced the incidence of verticillium wilt [32]. The significant reduction of *Verticillium dahliae* abundance in the BFN treatment as compared with the BFC treatment may be attributed to the biocidal potency of the mustard vegetable as a biofumigant. This study also agrees with Meng et al. [33] who reported that among different *Brassica* spp. tested, maceration of the plant tissue of mustard was highly effective in inhibiting *Fusarium oxysporum* growth in cucumber. The glucosinolate profile analysis showed that the *Brassica juncea* (Mustard) used in the BFN treatment contained a high amount of sinigrin (Table 3), which effectively kills soil-borne pathogens [34]. Some studies also showed that the nutrient surplus could explain the reduction of *Verticillium dahliae* abundance because the persistence of isothiocyanates in the soil is low [20,35]. Although there is a dearth of information on the comparison of biocidal potencies of various brassica vegetables with regards to disease suppression, glucosinolate profiles have been used as a basis for comparison [36,37]. This comparison, however, does not accurately give information on the efficacy level of brassica vegetables used relative to disease suppression and yield. The reduction in the abundance of *V. dahliae* in the BFN-treated soil translated to reduced disease incidence and better yield as compared to other treatments and the untreated control. This observation corresponds with findings from other studies that show that the use of brassica vegetables as biofumigant could also improve the yield of crops [38].

### 4.2. Biofumigation Could Change Soil Microbial Diversity and Composition

Analysis of the effects of treatments on the soil alpha diversity was done using the Shannon–Weaver diversity index. The bacterial and fungus mean values of the Shannon index in the BFN and BFC treatments were lower in the microbial diversity in the CF and CN treatments (Figure 1). The mean differences in the microbial species diversity index between the biofumigants suggest that mustard plant incorporation could better decrease the richness and diversity of microbes at the beginning of biofumigation. However, no statistical significance was observed at the fruiting stage, which suggests that the differences were overlapped by a homogenizing effect of the new selection pressure of the eggplant roots [39]. Further studies are needed to establish which one is the better biofumigant and the effects of other *Brassica* vegetables. Soil microbial diversity is an indicator of soil health, translating to better soil fertility and productivity. This study is consistent with earlier studies that suggest microbial diversity and richness correlate with the suppression of soil pathogens [40,41], which could result from antagonism and competition [42].

Nevertheless, a study by Hartman et al. [43] observed that the diversity of soil microbiota and the survival of pathogenic microbes were negatively correlated. The report clarified that the causal dynamics of diversity-infestation interactions might involve competition for the utilization of limited resources. The present study showed that the microbial diversity after biofumigation was enhanced, which could have altered the existing soil ecology and reduced soil-borne disease incidence in continuous eggplant cropping.

This more or less corresponds to the bacterial community structure detected in agricultural and other soil types [44,45]. Shen et al. [46] reported that the abundance of *Acidobacteria* in soil samples collected from disease-suppressive orchards improved compared to disease-conducive orchards. Noah et al. [47] also observed a higher abundance of *Acidobacteria* in potato common scab suppressive soils. Some studies showed that the long-term effects of biofumigation on the control of soil-borne plant pathogens are, to a large extent, caused by changes in the soil microbiome since this change could be triggered by nutrient addition after brassica plants are incorporated into the soil [18,20].

Our studies showed that the relative community abundance of *Pseudomonas*, *Flavobacter* and *Mortierella* in the BFN treatment was significantly higher than in the other treatments. The selected genera of both the microbial communities *Flavobacteria*, *Arthrobacter* and *Bacillus* in bacteria and *Chaetomium* and *Mortierella* in fungi are known to harbour strains that exhibit pathogen antagonism [48–50]. A similar result was obtained by Hollister et al. (2013) [51] who used mustard (a seed with fixed oil and GSLs), flax (a seed with fixed oil but no GSLs) and sorghum (contains neither) as controls to verify the effects of biofumigation. Their research results showed that mustard induced large increases in the abundance of bacterial taxa associated with fungal disease suppression (e.g., *Bacillus*, *Pseudomonas* and *Streptomyces* spp.). The phylotype richness of fungi decreased by >60% at the same time: all detected species (incl. *Alternaria*, *Fusarium*) showed reduced abundance. Other seeds (flax and sorghum) had less pronounced, mixed effects on fungi. The other reason behind this effect might be the higher tolerance of the biocontrol microbe against Brassica soil amendment decomposition products compared to phytopathogens, as shown in several papers [19,20]. This could likely have an inhibitory effect on the plant pathogen, which induced a disease index in the BFN treatment that was significantly lower than the other treatment. The observed trend showing an increase in the genera *uncultured_Anaerolineaceae* and a decrease in the others could be because of an increase in the anaerobic condition of the soils. Zhang et al. [52] reported that *Anaerolineaceae* increased with a manure ratio from 1.56% (0% manure) to 3.83% (25% manure), which could mean that higher SOM led to a lower oxygen supply, leading to an increased abundance of *Anaerolineaceae*. This could be the reason for the abundance in the BFN and BFC treatments because plant residues increase soil organic matter. It has been reported that *Anaerolineaceae* exist in anoxic soils and act as organic matter degraders [53]. With the development of the eggplant growth stage, the temporal difference in bacterial structure among different treatments was reduced, which could be caused by the changes in organic matter substrate availability. It could also be due to the growth stages of the plant since plant growth stages have been shown to affect microbial populations [54,55]. Although, there could be other effects indirectly caused by the host plant, which can also influence the conditions of the rhizosphere either physically or chemically. The exact mechanisms of how these factors interact with the rhizosphere community are still largely unknown. Further studies will be needed to elucidate other factors that could be responsible for the trend in this particular study.

*4.3. Biofumigation Change Soil Enzymatic Activities*

The incorporation of plants containing glucosinolates (GSLs) into soil results in its breakdown/hydrolysis by myrosinase enzymes into isothiocyanates (ITCs), which are capable of suppressing soil pathogens. It also has been reported that soil pathogens cannot reproduce under many nonhost crop rotation plants or are used as biofumigation materials [56]. The suppression observed is likely due to the effect of isothiocyanates released, following the hydrolysis of glucosinolate, that are present in biofumigation materials. Isothiocyanates are toxic to soil-borne pathogens and have been successfully used to reduce soil-borne populations of fungal pathogens [8]. The difference in the disease index of the biofumigation treatments could be due to the differences in the concentration and forms of the glucosinolate compounds released by the materials. It thus shows that the glucosinolate content of the *Brassica* crops influences the biocidal effect to some extent. The levels of verticillium wilt suppression varied because of the different types of glucosinolates, and this agrees with the study by Sarwar and Kirkegaard [57]. This finding is in line with former studies on the effects of

incorporating organic materials into the soil [58]. Friberg et al. [59] found variations in the microbial community structure following the incorporation of mustard.

Soil enzymes are derived from plant residues, and soil microbes [23] play major biochemical roles in the whole process of organic matter decomposition in the soil ecosystem. This study reported on two enzymes' urease and invertase. The soil enzymatic levels were higher in the BFN and BFC treatments compared to the other treatments. The addition of organic materials in the form of *Brassica* incorporation could have been responsible for increasing the activities of soil urease and invertase. This agreed with other reports that the addition of crop residues and organic amendments increased soil enzymes, soil humus content, enhanced soil microbial content and accelerated carbon and nutritional cycles [60]. Zhao et al. [61] also reported an improved urease activity following the application of a bio-organic fertilizer.

The experiment was conducted at a high latitude region where crops are cultivated once a year due to climate. Eggplant is usually cropped at the beginning of May till the end of August. After the eggplant is harvested, the temperature is still appropriate for planting the *Brassica* vegetable. Oilseed rape and mustard were planted in this experiment and were allowed to grow for 45 days at an optimum temperature of between 20–25 °C. After which, they were incorporated into the soil. This method could be regarded as an economically sustainable practice for the farmers in this region. Which, to some extent, can ameliorate their challenges with eggplant monocropping.

## 5. Conclusions

This study showed that crop rotation followed by biofumigation affected soil microbial structure, further controlling the soil pathogen of interest. *Brassica* containing high glucosinolate has the function of a biocidal effect on *V. dahliae* in the soil at the beginning of biofumigation, which induced a lower disease incidence rate and higher yield. The bacterial diversity of the BFN treatment was significantly lower than those of the other treatments, and beneficial bacterial community relative abundance increased after the soil was biofumigated for 15 days. However, the temporal difference in bacterial structure among the different treatments was reduced with the development of the eggplant growth stage. In summary, biofumigation using mustard was efficient in controlling eggplant *Verticillium* wilt due to changes in the soil bacterial composition at the early eggplant growth stage, and initial conditions in the soil bacterial community are a key determinant of what is going to happen after *Brassica* biofumigation, which could be considered a practical addition to integrated management for the reduction of soil-borne pathogens.

**Author Contributions:** L.M.: Experiment design, investigation, sampling and analysis, funding acquisition. Y.Z.: Writing—original draft preparation. S.Y.: Writing—reviewing. A.O.O.: Writing—original draft. S.Z.: Data analysis. S.L.: Editing, project administration, supervision, funding acquisition. All authors have read and agreed to the published version of the manuscript.

**Funding:** This research was funded by the Natural Science Joint Guidance Foundation of Heilongjiang Province, China grant number (LH2019C041), and Sinograin II: technological innovation to support environmentally friendly food production and food safety under a changing climate—opportunities and challenges for Norway–China cooperation (CHN-17/0019).

**Institutional Review Board Statement:** Not applicable.

**Informed Consent Statement:** Not applicable.

**Data Availability Statement:** Most of the collected data are contained in the tables and figures in the manuscript.

**Conflicts of Interest:** The authors declare no conflict of interest.

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
