# Peer review of "Temporal Assessment of Biofumigation Using Mustard and Oilseed Rape Tissues on Verticillium dahliae, Soil Microbiome and Yield of Eggplant"

_agronomy, doi:10.3390/agronomy12122963_

Round 1
Reviewer 1 Report
Please open the attached file.

Author Response
Responses to the comments
Manuscript Number: Agronomy-2028402
We thank you and the reviewers for the very helpful comments on our manuscript ‘Temporal Assessment of Bio-Fumigation Using Mustard and Oilseed Rape Tissues on Verticillium Dahliae, Soil Microbiome and Yield of Eggplant (Agronomy-2028402)’. We have carefully considered all comments and thoroughly revised the text. We also have prepared all of the items required by the submission system. We hope the revised manuscript is suitable for publication in Agronomy.
All our responses to reviewers are detailed in blue. Changed made in the manuscript are highlighted. The revision has been developed in consultation with all coauthors, and each author has given approval to final form of this revision.
Prof. Shumin Li
Resource and Environmental College, Northeast Agricultural University
Reviewer #1: Agronomy-2028402
The aim of the research presented in this manuscript was to aim of this study is of the study were (i) to evaluate the effects of bio-fumigation on verticillium wilt disease of eggplant and the yield of eggplant (ii) to compare the effects of different glucosinolate concentrations containing crops in the bio-fumigants used, and (iii) to assess the temporal effects of the bio-fumigation materials on the microbial community structures. In terms of methodology, the research is carried out correctly and the results are presented very clearly. The chapter Discussion has been written in an interesting way. The summary is factual. Before that, however, this manuscript needed some corrections.
All the comments are valuable. We accepted the changes and thoroughly revised the whole text including the English according to the comments. All our responses are detailed in blue. The line number in the response to every comment as follows was that in the revised manuscript with “No changes marked”.
[1] In the introduction section, the author should provide a novelty statement at the end. What new things have authors done or correlated in this research compared to old ones?
Response: We clarified the novelty and aims of our research at the end of introduction. (line 69-71,line 75-78)
[2] The systematic abstract is missing. Introduce the need for study in 1-2 lines. Then please give a clear-cut point problem source as a problem statement that is tackled in the current study. Also, give a logical reason for selecting the current strategy or treatments. Then provide a definitive conclusion withdrawn through research in a single line.
Response: We have revised the abstract according the suggestions. (line 11-13, line 11-13,line 31-34) in abstract.
[3] Give a logical reason for selecting the current strategy, i.e., Bio-Fumigation Using Mustard and Oilseed Rape Tissues on Verticillium Dahliae.
Response: The reason why we selected the Mustard and Oilseed Rape Tissues as biogumigant and the current strategy was clarified in introduction part. (line 75-78)
[4] The authors should follow the title in the introduction section, i.e., Temporal Assessment of Bio-Fumigation Using Mustard and Oilseed Rape Tissues on Verticillium Dahliae , Soil Microbi-3 ome and Yield of Eggplant. Do you consider the topic original or relevant in the field? Does it address a specific gap in the field?
Response: The instruction has been revised according the suggestions. (Page 2 line 65-71,line75-78)
[5] Keywords should be in alphabetical order and should not duplicate words appearing in the title.
Response: We have modified the keywords in alphabetical order. (Page 1 line 35)
[6] Page 1 line 3: should add space between two words “management” and “for” in the following words “managementfor”
Response: Done (line 34, line 427)
[7] Page 1 line 24: change the “biofumiageted” to “biofumigated”
Response: Done (line 26)
[8] Page 1 line 24: Change “of” into “in” in the following sentence “the temporal difference of bacterial structure” and at whole the manuscript.
Response: Done (line 27, line 370)
[9] Page 1 line 44: Remove the extra space that located before the word “Generally,”
Response: Done (line 46)
[10] Page 3 line 100: change the format of date from “on May 6, 2016, soil samples” to (on 6th of May, 2016)
Response: Done (line 113)
[11] Page 3 line 103, 104: Remove the under line from the symbol of degree “-4oC” and at whole of manuscript
Response: Thanks very much for your comments and suggestions to improve the quality of this manuscript. According to your description, we have found the corresponding position, but this is not our problem, it is caused by the font selection of the journal. (line 117)
[12] Page 3 line 104: add the full stop at the end of the following paragraph “ultra-low temperature freezer”
Response: Done (line 117)
[13] In the section of statistical analysis, the authors should clarify the type of Experimental design (RCBD, RCD) and kind of ANOVA (One-Way or Two-Way) and how many biological samples have been taken from soil or plants during experimentations. Also, the author should mentioned the type of correlation has been used “Bray-Curtis dissimilarities” and which program has been used to perform it.
Response: Thanks very much for your comments and suggestions to improve the quality of this manuscript. We clarified the experimental design (RCBD)in our Materials and methods (Page 3 line 99-100). For the ANOVA test, we have conducted a detailed supplement to this test as a One-Way ANOVA test. The NMDS analysis in this study was calculated using the Bray-Curtis dissimilarity matrix from the OTU table, which assessed soil bacteria and fungi communities and structures. (line 181-184)
[14] Page 4 line 165 remove the extra space between two words “done by” and “analysis”
Response: Done (line 181)
[15] The authors also should mentioned what is indicating the bar in the figures (standard deviation or standard error) in the legends
Response: Thanks very much for your comments and suggestions to improve the quality of this manuscript. We have modified the figure 3 and added explain of the bar in the legends. (Figure 3).
[16] Page 5 line 190: remove the extra space between “index” and “in BFN”
Response: Done (line 208)
[17] Page 5 line 192 and page 6 lines 207 and 208: Remove the extra brackets
Response: Done. Please see line 211, line 226, line 228.
[18] The authors should be mentioned what indicates the number 1 in the following values “35.04±2.9a1 and 0.33±0.002c1 “ in the table 1 and 3?
Response: Thanks very much for your comments and suggestions to improve the quality of this manuscript. We have explained this in the notes below the table, and the details can be found in the table 1 and table 3.
[19] The conclusion is so much descriptive. Please provide a conclusive conclusion, Add the targeted beneficiary audience who will get benefit from this research. Also, give clear-cut recommendations Give future prospective regarding this research.
Response: Thanks very much for your comments and suggestions to improve the quality of this manuscript. We revised the conclusions and add “bio-fumigation using mustard was efficient in controlling eggplant Verticillium wilt due to changing soil bacterial composition of early eggplant growth stage and initial conditions in soil bacterial community is a key determinant of what is going to happen after a Brassica biofumigation” which we got the conclusive conclusion at present study. (line 423-427)
[20] Update the old references (citations) especially numbers from 3,5,13,16,18,29,20 and 50
Response: We updated the all above references and add some new references (See reference:3,5, 19,20,21, 23, 25, 51)

Reviewer 2 Report
Dear Authors,
I think your paper requires revisions. Please address the following points, detailed below.
Major issues
1., The novelty of the data presented in the material is unclear. While you mention [25] as an example, some other studies (Garibaldi et al., 2010; Inderbitzin et al., 2018; Ogundeji et al., 2022) already tested Brassica biofumigation in Verticillium disease in eggplant. Compare your results to theirs in a critical part of the discussion. One of these is your just published study. Why is it not mentioned in the paper?
2. In my opinion, the discussion needs points to be added, given the fact that several pieces of relevant literature exists.
2a., The possible differences between the two Brassica biofumigants can be the following: total GSL amount is much higher for mustard (though also see my point 4), the volatility therefore the penetration capacity of the allyl isothiocyanate from sinigrin in mustard is greater than that of the ITCs from the other Brassica, the differences in the ability to confer effects via nutrient addition hence changing the abundance of antagonistic organisms, etc. Several issues relevant to your discussion were recently reviewed in sections 6.3., 6.5.-6.8. of (Plaszkó et al., 2021). I also suggest instead of linking to studies with manure (L328), consult studies which used Brassica (and sometimes compared to non-Brassica!) seed meals, e.g. (Hollister et al., 2013; Hu et al., 2019).
2b., L302-307: From various studies, no one-scheme-fits-all conclusion could be drawn. I think it might be wise to speculate that initial conditions in the community is a key determinant of what is going to happen after a Brassica biofumigation treatment. See e.g. section 6.7 of (Plaszkó et al., 2021) and references therein for various outcomes of Brassica and ITC biofumigation on soil microbial communities.
3., Why was the protocol in L106-109 necessary? It seems you added a major modification to the manufacturer protocol on purpose, what is more, it looks potentially harmful for the DNA in the sample. Add a reference to support the decision.
4., GSL determination. The method for GSL determination ISO 9167-1:1992 has as withdrawn status, possibly due to having become obsolete. Why was this method chosen? You cannot use sinigrin as internal standard, it is present in mustard. There is no mention of using authentic standards for “external calibration” – perhaps that is what you meant under “internal standard”, please clarify. Note it is not trivial to use sinigrin to obtain equivalents of other GSL classes. What standards were used, what purity? What was the linear range of determination under your conditions? Column dimensions are missing. Altogether, your method must be shown to be producing valid concentration values.
5., Are soil data given for DW? Please clarify (e.g. gene count).
Minor issues
L15: abstract says for 12 years, in M&M, it is >13 years. Which one is correct?
L17, L30 and elsewhere: taxonomic names (genera, species) should be in italic
L25 and elsewhere: the manuscript contains many double spaces: “ “
L27-28 and elsewhere: unclassified species should not be written as such in the abstract. In the full text, OTU_otu-number might be OK, but in the abstract and when talking about the OTU, “an unidentified fungus” should be used.
L82: please double-check that the provided P and K data are not P2O5 and K2O equivalents.
L84: Brassica
L132: data
L132: Demultiplexed
L134: USEARCH
L140: I guess there is a typo here, the fruit setting period and the final fruit period is separated by a few days only
L149: mustard
L219: ten times
L301: the beginning of biofumigation
L320: abundance
Figure 3 does not contain any error bars. Its resolution is also very poor.
Consider using the term “benzenic” GSLs instead of “aryl” GSLs. “Aryl” or “Aromatic” might also include the indolic GSLs depending on context.
There might be additional typos, though I listed several ones above. Please check the paper with a word processor dictionary module.
The three figures show three styles. Please make them uniform.
Best regards.
References
Garibaldi, A., Gilardi, G., Clematis, F., Gullino, M. L., Lazzeri, L., & Malaguti, L. (2010). Effect of green Brassica manure and Brassica defatted seed meals in combination with grafting and soil solarization against Verticillium wilt of eggplant and Fusarium wilt of lettuce and basil. Acta Horticulturae, 883, 295–302. Scopus. https://doi.org/10.17660/ActaHortic.2010.883.36
Hollister, E. B., Hu, P., Wang, A. S., Hons, F. M., & Gentry, T. J. (2013). Differential impacts of brassicaceous and nonbrassicaceous oilseed meals on soil bacterial and fungal communities. FEMS Microbiology Ecology, 83(3), 632–641. Scopus. https://doi.org/10.1111/1574-6941.12020
Hu, P., Wu, L., Hollister, E. B., Wang, A. S., Somenahally, A. C., Hons, F. M., & Gentry, T. J. (2019). Fungal Community Structural and Microbial Functional Pattern Changes After Soil Amendments by Oilseed Meals of Jatropha curcas and Camelina sativa: A Microcosm Study. Frontiers in Microbiology, 10. https://doi.org/10.3389/fmicb.2019.00537
Inderbitzin, P., Ward, J., Barbella, A., Solares, N., Izyumin, D., Burman, P., Chellemi, D. O., & Subbarao, K. V. (2018). Soil microbiomes associated with verticillium wilt-suppressive broccoli and chitin amendments are enriched with potential biocontrol agents. Phytopathology, 108(1), 31–43. Scopus. https://doi.org/10.1094/PHYTO-07-17-0242-R
Ogundeji, A. O., Meng, L., Cheng, Z., Hou, J., Yin, T., Zhang, S., Liu, X., Liu, X., & Li, S. (2022). Integrated crop practices management stimulates soil microbiome for Verticillium wilt suppression. European Journal of Agronomy, 140. Scopus. https://doi.org/10.1016/j.eja.2022.126594
Plaszkó, T., Szűcs, Z., Vasas, G., & Gonda, S. (2021). Effects of glucosinolate-derived isothiocyanates on fungi: A comprehensive review on direct effects, mechanisms, structure-activity relationship data and possible agricultural applications. Journal of Fungi, 7(7), Article 7. https://doi.org/10.3390/jof7070539
Author Response
Responses to the comments
Manuscript Number: Agronomy-2028402
We thank you and the reviewers for the very helpful comments on our manuscript ‘Temporal Assessment of Bio-Fumigation Using Mustard and Oilseed Rape Tissues on Verticillium Dahliae, Soil Microbiome and Yield of Eggplant (Agronomy-2028402)’. We have carefully considered all comments and thoroughly revised the text. We also have prepared all of the items required by the submission system. We hope the revised manuscript is suitable for publication in Agronomy.
All our responses to reviewers are detailed in blue. Changed made in the manuscript are highlighted. The revision has been developed in consultation with all coauthors, and each author has given approval to final form of this revision.
Prof. Shumin Li
Resource and Environmental College, Northeast Agricultural University
Reviewer #2: Agronomy-2028402
I think your paper requires revisions. Please address the following points, detailed below.
Thanks for all helpful comments. We revised the whole text accordingly. The line numbers in the response to every comment as follows are those in the revised manuscript with changes marked.
Major issues
[1] The novelty of the data presented in the material is unclear. While you mention [25] as an example, some other studies (Garibaldi et al., 2010; Inderbitzin et al., 2018; Ogundeji et al., 2022) already tested Brassica biofumigation in Verticillium disease in eggplant. Compare your results to theirs in a critical part of the discussion. One of these is your just published study. Why is it not mentioned in the paper?
Response: Thank you very much for reviewer’s useful references. We read all references carefully. We clarified the novelty of this paper in the end of introduction (Page 2 line 65-71 there is a dearth of information on bio-fumigation of crops using plant tissues of different glucosinolate levels, their suppressive effects on Verticillium dahliae and the consequent temporal assessment variation in soil microbial composition; and soil enzyme activities, especially in continuous eggplant cropping systems.) The research of Garibaldi et al. (2010) is about control effects of eggplant Verticillium Wilt, but it didn’t have results of soil microbial. Inderbitzin et al. (2018) contain soil microbial results,but they used Broccoli as biofumigation material. We use another two kinds of brassica plant (mustard and oilseed Rape) that contain two levels content of glucosinolate we screened, which growth period is short and suitable for rotation with eggplant in our area. Our paper published in 2022(Ogundeji et al., 2022)is focus on biochar and grafted’s effects on control the Verticillium Wilt of eggplant according to the results of biofumigation. Its content is different from this manuscript.
We updated these references in introduction and discussion parts(Ref:3,10,17,18,20,38,51)and clarify the novelty of this paper at the end of in introduction. (line 69-71,line 79-82)
[2] In my opinion, the discussion needs points to be added, given the fact that several pieces of relevant literature exists.
[2a] The possible differences between the two Brassica biofumigants can be the following: total GSL amount is much higher for mustard (though also see my point 4), the volatility therefore the penetration capacity of the allyl isothiocyanate from sinigrin in mustard is greater than that of the ITCs from the other Brassica, the differences in the ability to confer effects via nutrient addition hence changing the abundance of antagonistic organisms, etc. Several issues relevant to your discussion were recently reviewed in sections 6.3., 6.5.-6.8. of (Plaszkó et al., 2021). I also suggest instead of linking to studies with manure (L328), consult studies which used Brassica (and sometimes compared to non-Brassica!) seed meals, e.g. (Hollister et al., 2013; Hu et al., 2019).
Response: We discussed effects of nutrient addition on changing the abundance of antagonistic organisms in discussion part. (line 307-309, line 343-346)
We also discussed results of change of soil microorganism affected by Brassica seed meals and non-Brassica (line 335-338). We updated the related reference (Plaszkó et al., 2021). (Page 10 line 307-309, Page 10 line 343-346)
[2b] L302-307: From various studies, no one-scheme-fits-all conclusion could be drawn. I think it might be wise to speculate that initial conditions in the community is a key determinant of what is going to happen after a Brassica biofumigation treatment. See e.g. section 6.7 of (Plaszkó et al., 2021) and references therein for various outcomes of Brassica and ITC biofumigation on soil microbial communities.
Response: We revised the conclusion according reviewer’s comment. (line 423-427)
[3] Why was the protocol in L106-109 necessary? It seems you added a major modification to the manufacturer protocol on purpose, what is more, it looks potentially harmful for the DNA in the sample. Add a reference to support the decision.
Response: This is a method for pretreatment of soil DNA extraction, which is repeated 3-5 times through liquid nitrogen quick freezing and bath at 65-70℃, in order to promote the breaking of soil microbial cell wall. It is not harmful for DNA in the sample because time is very short. The following reference was added in M&M. (line 121)
Wu, M.; Zhang, H.; Li, X.; Su, Z.; Zhang, C. An extraction method of fungal DNA from soils in North China. Chinese Journal of Ecology 2007, 26, 611-616.
[4] GSL determination. The method for GSL determination ISO 9167-1:1992 has as withdrawn status, possibly due to having become obsolete. Why was this method chosen? You cannot use sinigrin as internal standard, it is present in mustard. There is no mention of using authentic standards for “external calibration” – perhaps that is what you meant under “internal standard”, please clarify. Note it is not trivial to use sinigrin to obtain equivalents of other GSL classes. What standards were used, what purity? What was the linear range of determination under your conditions? Column dimensions are missing. Altogether, your method must be shown to be producing valid concentration values.
Response: The method we used for GSL determination is according the method we published in 2007. We added this reference in M&M part. (line 164).
Shumin Li, Ilona Schonhof,Angelika Krumbein, Long Li,Hartmut Stützel, Monika Schreiner, Glucosinolate Concentration in Turnip (Brassica rapa ssp. rapifera L.) Roots as Affected by Nitrogen and Sulfur Supply, J. Agric. Food Chem. 2007, 55, 8452–8457.
Sinigrin was bought from Sigma company. Its purity is 99%. Each sample was duplicated two times. One of the duplicates was added two hundred microliters of a 5 mM (2077 g/L) stock solution of sinigrin in methanol just before the first extraction as internal standard. A Spherisorb ODS2 column (5 µm, 250 × 4 mm)(line170)
was used to measure to GLS. The GLS concentration was calculated using sinigrin as internal standard and the response factor of each compound relative to sinigrin. We add GLS measurement method in M&M part. (line165-169)
[5] Are soil data given for DW? Please clarify (e.g. gene count).
Response: Yes. Number of V. dahlia was expressed by gene copied g-1DW soil. We revised it. (line 131) and table 1.
Minor issues
[1] L15: abstract says for 12 years, in M&M, it is >13 years. Which one is correct?
Response: The fumigation was carried out in 12 years of continuous crop soil, the samples measured were in the 13th year thirteen, we revised it in them manuscript. (Page 1 line 17)
[2] L17, L30 and elsewhere: taxonomic names (genera, species) should be in italic
Response: Done (Page 1 line 19)
[3] L25 and elsewhere: the manuscript contains many double spaces: “ “
Response: We checked the entire text exactly as given and corrected the same errors therein.
[4] L27-28 and elsewhere: unclassified species should not be written as such in the abstract. In the full text, OTU_otu-number might be OK, but in the abstract and when talking about the OTU, “an unidentified fungus” should be used.
Response: Thanks very much for your comments and suggestions to improve the quality of this manuscript. We did not find the use of the "OTU-number" representation in the summary section upon closer inspection. We deleted the unclassified species in abstract. (line 26-29)
[5] L82: please double-check that the provided P and K data are not P2O5 and K2O equivalents.
Response: We have modified P and K data as P2O5 and K2O equivalents. (Page 2 line 95-96)
[6] L84: Brassica (Page 3 line 97)
Response: Done.
[7] L132: data
Response: Done. (Page 4 line 146)
[8] L132: Demultiplexed
Response: Done. (Page 4 line 146)
[9] L134: USEARCH.
Response: Done. (Page 4 line 148)
[10] L140: I guess there is a typo here, the fruit setting period and the final fruit period is separated by a few days only
Response: The description here is correct, because eggplant is a continuously harvested crop, so we picked fruit of eggplant continuously from June 29th to August 4th measure the yield. (line 155)
[11] L149: mustard
Response: Done (Page 4 line 163)
[12] L219: ten times
Response: Done (Page 7 line 238)
[13] L301: the beginning of biofumigation
Response: Done (Page 10 line 324)
[14] L320: abundance
Response: Done (Page 11 line 350)
[15] Figure 3 does not contain any error bars. Its resolution is also very poor.
Response: Thanks very much for your comments and suggestions to improve the quality of this manuscript. We redidfig.2 and fig.3, and added the error bar in fig.3 accordingly. (Page 8 line 246-249)
[16] Consider using the term “benzenic” GSLs instead of “aryl” GSLs. “Aryl” or “Aromatic” might also include the indolic GSLs depending on context.
Response: Thanks very much for your comments and suggestions to improve the quality of this manuscript. We have revised “benzenic” GSLs instead of “aryl” GSLs the whole text.
[17] There might be additional typos, though I listed several ones above. Please check the paper with a word processor dictionary module.
Response: We checked the entire text exactly as given and corrected the same errors there in.
[18] The three figures show three styles. Please make them uniform.
Response: We redid the Fig.1,Fig.2 and fig.3 and made them uniform.

Round 2
Reviewer 1 Report
The authors have been did all the correction and replied well on all comments so i recommended publishing the manuscript in Agronomy journal
Author Response
Dear Reviewer,
We thank you for your approval of our manuscript modification‘Temporal Assessment of Bio-Fumigation Using Mustard and Oilseed Rape Tissues on Verticillium Dahliae, Soil Microbiome and Yield of Eggplant (Agronomy-2028402)’. Your comments are very helpful in improving our manuscript quality. Thank you again for your patience in reviewing our paper and we wish you a happy life.
with best wishes,
Shumin Li*, Lingbo Meng, Yuhang Zhang, Shaopeng Yu, Abiola O. Ogundeji, Shu Zhang

Reviewer 2 Report
Dear Authors,
Thank you for the revised version and the answers to my queries. I’m now comfortable with the novelty issue, and I also think the discussion revision is OK (previous points 1-2). I note that the robustness freeze – thaw cycle used to extract DNA heavily depends on the activity of DNAses in the sample (previous point 3).
Though your answer to previous point 2a links to the acknowledgement (L423-427), I think I found the part what you wanted to link to.
The following points are left to address, in my opinion.
1. (previous point 4)., I now understand that the usage of internal standards was actually a spiking procedure, which is OK to use. Again, this M&M section must contain enough details so that one is convinced that you’re producing valid concentration values – this is still not true for GSLs other than sinigrin.
1a., The used range of linear determination (calibration curve) should be noted.
1b., In addition, there is still no mention of using authentic standards for calibration, other than sinigrin, whereas you report exact concentrations for a wide range of GSLs. In L166-167 you mention authentic standards, these have to be explicitly listed with manufacturer and purity.
1c., L165: 5 mM is not 2077 g/L (the latter is not even physically meaningful), please revise.
2., L320-322: or, alternatively, the differences were overlapped by a homogenizing effect of the new selection pressure, the eggplant roots
3., L217: NMDS is a distance-metric and the plot is unlike that of PCA. It is not the distribution, but the relatively high distance among the centres of the clusters and the relatively low distance between the clusters that show the fact you (correctly) claim.
Minor issues.
L287: ITCs are the default decomposition products
L387: no reference number for Sarwar and Kirkegaard (1998)
Typos, English.
L11: is a soil-borne pahtogen that is difficult to eliminate
L18: was used
L18: PCR was used to assay
L29: Chaetomium genera were significantly
L42 Kleb is an author name, it should not be in italic
L59: severly reduced the growth of (?)
L59: few studies
L105: revise, this is unclear (what do you mean by “fresh”?)
L117: soil samples weighing approximately 500 mg were placed
Table 1 header: copies g-1
L346: obtained by [51] when using
L393: urease and invertase (non-capital letters)
L395: Brassica (italics)
L404: mustard were planted
Best regards.
Author Response
Responses to the comments
Manuscript Number: Agronomy-2028402
We thank you and the reviewers for the very helpful comments on our manuscript ‘Temporal Assessment of Bio-Fumigation Using Mustard and Oilseed Rape Tissues on Verticillium Dahliae, Soil Microbiome and Yield of Eggplant (Agronomy-2028402)’. We have carefully considered all comments and thoroughly revised the text. We also have prepared all of the items required by the submission system. We hope the revised manuscript is suitable for publication in Agronomy.
All our responses to reviewers are detailed in blue. Changed made in the manuscript are highlighted. The revision has been developed in consultation with all coauthors, and each author has given approval to final form of this revision.
Prof. Shumin Li
Resource and Environmental College, Northeast Agricultural University
Reviewer #2: Agronomy-2028402
Thank you for the revised version and the answers to my queries. I’m now comfortable with the novelty issue, and I also think the discussion revision is OK (previous points 1-2). I note that the robustness freeze – thaw cycle used to extract DNA heavily depends on the activity of DNAses in the sample (previous point 3).
Though your answer to previous point 2a links to the acknowledgement (L423-427), I think I found the part what you wanted to link to.
The following points are left to address, in my opinion.
Thanks for all helpful comments. We revised the whole text accordingly. The line numbers in the response to every comment as follows are those in the revised manuscript with changes marked.
- (previous point 4)., I now understand that the usage of internal standards was actually a spiking procedure, which is OK to use. Again, this M&M section must contain enough details so that one is convinced that you’re producing valid concentration values – this is still not true for GSLs other than sinigrin.
1a., The used range of linear determination (calibration curve) should be noted.
Response: The glucosinolate concentration was calculated using the response factor of each compound relative to sinigrin. The method was set up by Institute of Vegetable and Ornamental Crops, Grossbeeren/Erfurt e.V., Germany when we did Sino-German cooperation project.
1b., In addition, there is still no mention of using authentic standards for calibration, other than sinigrin, whereas you report exact concentrations for a wide range of GSLs. In L166-167 you mention authentic standards, these have to be explicitly listed with manufacturer and purity.
Response: We don’t need to have every pure individual glucosinolate composotion. We used mass spectrum to identify individual glucosinolates in standard reference materials of oilseed rape that enriched glucosinolate composition. Individual glucosinolate in biofumigation materials were identified by comparison of its retention time with that of individual glucosinolate in standard reference materials. See line 177-183.
1c., L165: 5 mM is not 2077 g/L (the latter is not even physically meaningful), please revise.
Response: Yes. 2077 g/L is not even physically meaningful. It was deleted. See line 167.
2., L320-322: or, alternatively, the differences were overlapped by a homogenizing effect of the new selection pressure, the eggplant roots
Response: Response: We changed the sentence “this sampling period could have caused dynamic shifts in microbial communities”to “the differences were overlapped by a homogenizing effect of the new selection pressure of the eggplant roots.” See line 330-331.
3., L217: NMDS is a distance-metric and the plot is unlike that of PCA. It is not the distribution, but the relatively high distance among the centres of the clusters and the relatively low distance between the clusters that show the fact you (correctly) claim.
Response: Although the calculation methods of PCA and NMDS are different, results of their analysis are similar. PCA and NMDS both use the distance between points to show the similarity or difference of samples. NMDS can convert distance algorithm, and PCA is a European distance algorithm. PCA algorithm is implemented as numerical sorting, and NMDS algorithm is implemented as ranking order.
We changed the“distributed”to “ were separated”. See line 225.
Minor issues.
L287: ITCs are the default decomposition products
Response: Yes. I think the main decomposition product of glucosinolate that has the function of disinfection is ITCs.
We inserted “main” before “isothiocyanates”. See line 295.
L387: no reference number for Sarwar and Kirkegaard (1998)
Response: The reference number was added. See line 567.
Typos, English.
L11: is a soil-borne pahtogen that is difficult to eliminate
Response: Done. See line 11.
L18: was used
Response: Done. See line 18.
L18: PCR was used to assay
Response: Done. See line 18.
L29: Chaetomium genera were significantly
Response: Done. See line 29.
L42 Kleb is an author name, it should not be in italic
Response: Done. See line 43.
L59: severly reduced the growth of (?)
Response: Done. See line 59.
L59: few studies
Response: Done. See line 70.
L105: revise, this is unclear (what do you mean by “fresh”?)
Response: It has been revised. See line 107.
L117: soil samples weighing approximately 500 mg were placed
Response: It has been revised. See line 119.
Table 1 header: copies g-1
Response: It has been revised. See table 1.
L346: obtained by [51] when using
Response: It has been revised. See line 356.
L393: urease and invertase (non-capital letters)
Response: Done. See line 403.
L395: Brassica (italics)
Response: Done. See line 405.
L404: mustard were planted
Response: Done. See line 414.
